# The Effects of a Multifunctional Rust Inhibitor on the Rust Resistance Mechanism of Carbon Steel and the Properties of Concrete

Zhiqiang Niu *, Xiaoming Lu and Yanan Luo

Faculty of Civil and Architectural Engineering, Zhengzhou University of Science & Technology, Zhengzhou 450064, China; luxiaoming2010@zit.edu.cn (X.L.); luoyn@zit.edu.cn (Y.L.)
* Correspondence: nzq815@zit.edu.cn

**Abstract:** To address rebar corrosion in existing concrete structures, a multifunctional compound rust inhibitor was developed. This study investigates the impact of this inhibitor on carbon steel rust resistance, as well as the mechanical properties and durability of concrete. The results demonstrate a significant reduction in weight loss of carbon steel when using a multifunctional rust inhibitor, with a rust inhibition efficiency of 82.6%. Scanning electron microscopy (SEM) and atomic force microscopy (AFM) were employed to observe and analyze the surface elements of carbon steel, both with and without the presence of a rust inhibitor. The findings indicate that the use of a rust inhibitor leads to a smoother and more stable surface film. The results of the experiments on compressive strength, chloride ion electromobility, and the rapid carbonation test of concrete with varying rust inhibitor contents indicate that increasing the amount of rust inhibitor can improve the compressive performance of concrete and can regulate the mobility of chloride ions. Specifically, when the rust inhibitor content reaches 4%, it has a notable positive impact on the performance of concrete, and further increases in content lead to smoother results.

**Keywords:** corrosion inhibitor; carbon steel; concrete; antirust mechanism

## 1. Introduction

Concrete material has become a widely used building material because it is easy to cast, wear resistance, and low cost; it is also the first choice of major infrastructure construction projects. For most important construction projects with huge investments, mostly in the form of reinforced concrete, their service life is generally required to reach 100 years [1]; however, the service life of many construction projects is seriously reduced due to corrosion problems [2,3]. This is especially true for construction projects located in underground and marine environments where corrosive substances are abundant. Chloride ion erosion has been identified as the primary factor that leads to a reduction in the durability of reinforced concrete structures. During this process, chloride ions in the environment enter the concrete from the surface through capillary action, osmosis, diffusion, and physical and chemical adsorption, and therefore penetrate from the protective layer to the reinforcement surface, resulting in passivation film destruction and pore erosion [4–7]. Under the action of air and water, reinforced concrete components form a corrosion potential difference, inducing steel corrosion [8]. Further rust expansion can damage the concrete protective layer, which in turn weakens the bearing capacity and integrity of reinforced concrete structures. Hence, enhancing the anti-corrosion performance of reinforced concrete components is crucial for improving the overall durability of concrete structures [9].

Methods to suppress steel corrosion in reinforced concrete structures, in general, include electrochemical chlorine removal, corrosion resistant steel, realkalization, and steel rust inhibitor. Among these methods, the more simple, convenient and economic method is to add a steel rust inhibitor as an admixture to inhibit steel corrosion. In a material

system composed of steel bar and concrete, adding a small amount of substance can effectively delay the occurrence of corrosion and can reduce the corrosion rate of steel bar; this substance is called a rust inhibitor [10,11]. Under the condition of chloride ion erosion, a steel bar rust inhibitor has two roles; one role is to delay the corrosion time of the steel bar by using the critical chloride ion coefficient, and the other role is to slow down the corrosion speed of the steel bar [12–14]. Rust inhibitors, according to their chemical composition, can be divided into inorganic-type, organic-type, and mixed-type rust inhibitors. Inorganic rust inhibitors mainly include nitrite, nitrate, chromic salt, phosphate, and so on. Nitrite is the world's earliest application of rust inhibitor, and with the most obvious effect as a rust inhibitor, it is also widely used in reinforced concrete structure as a rust inhibitor [15,16]. Studies conducted by RAJA [17] in the 1970s on calcium nitrite showed that it had good rust resistance and it had no negative effects on concrete. Ramasu et al. [18] showed that calcium nitrite rust inhibitor could effectively inhibit steel corrosion in alkaline solutions containing chloride ions, and its protective effect was related to the concentration ratio of $NO_2^-$ and $Cl^-$, which had a critical value threshold. Further increases in nitrite concentration could not further reduce the corrosion rate of reinforcement. Yi, B. et al. [19] found that the addition of nitrite rust inhibitor thickened the passivation film on the surface of carbon steel and accelerated the growth rate of the passivation film, thus delaying the corrosion of carbon steel. Tang, Y.B. et al. [20] studied the effects of nitrite and organic amine inhibitors on carbon steel passivating film in a simulated concrete hole solution containing chlorine salts by using electrochemical noise comparison, and found that the same amount of nitrite could repair metastable holes faster than organic amine inhibitors.

The above research shows that nitrite is a type of rust inhibitor, with good performance as an economic steel bar rust inhibitor, that does not cause a negative impact on concrete, and it is the most widely used in engineering [21–24]. However, due to the toxicity of nitrite, it is a compound that needs to be used with other materials, which can reduce the amount and ensure its rust resistance effect [25]. Oliveira et al. [26] combined sodium nitrite with benzotriazole as a cathode rust inhibitor, and the rust inhibition effect obtained was more stable than that obtained by using a single rust inhibitor. Simović, A.R. et al. [27] used nitrite as the main component and mixed a small amount of D-gluconate sodium to make a compound rust inhibitor. Through an electrochemical test, it was found that in the concrete simulation hole solution with pH = 11.0, 0.02 mol/L $NaNO_2$ and 0.01 mol/L D-gluconate sodium obtained the best rust inhibition effect. The disadvantages of using $NaNO_2$ alone can be overcome. In addition, when nitrite ions are mixed into concrete, only the free nitrite in the concrete pore liquid can be transferred to the surface of the steel bar, repair the passivation film, and achieve the effect of inhibiting the corrosion of the steel bar [15,28]. The nitrite solution in the process of transport is often part of the adsorption by the nanopore solidification and can not play the role of rust inhibition. Therefore, investigating the ionization and adsorption of nitrite ions in the process of nanopore transmission plays a key role in the final rust inhibition effect of a nitrite inhibitor in concrete.

In this work, according to previous research results, a multifunctional rust inhibitor was prepared by choosing molybdate and nitrite as the main components. The corrosion rate and rust resistance efficiency of carbon steel samples were calculated using a weight loss test under different multifunctional compound rust inhibitor dosages. SEM and AFM were used to observe the surface of carbon steel soaked in the solution with and without rust inhibitor. The influence of the rust inhibitor on the surface of carbon steel was studied and the mechanism of the rust inhibitor was further analyzed. The multifunctional rust inhibitor was mixed into concrete with a certain mass fraction; the compressive strength, chloride ion diffusion coefficient, carbonation experiment, and electrochemical test of the concrete were tested; the influence of the rust inhibitor on the performance of concrete was discussed; and the appropriate dosage of the multifunctional rust inhibitor in concrete was put forward.

## 2. Experiments

### 2.1. Materials Preparation

The cement used in the experiments is P·O 42.5 ordinary Portland cement; the physical properties of the cement are shown in Table 1 and the chemical properties of the cement are showed in Table 2. The test results showed that the cement had good physical properties, and the related indexes met the basic GB 175-2007 "General Purpose Portland Cement" specifications for concrete. The sand used was standard sand, in line with ISO R679-68 standards; the particle size ranged from 0.08 mm to 2.0 mm, the mud content was less than 0.2%, and the $SiO_2$ content was greater than 96%.

**Table 1.** Physical properties of the cement.

| Density (kg/m³) | Specific Surface Area (kg/m²) | Normal Consistency (%) | Time of Coagulation (min) | | Compressive Strength (3d, MPa) |
|---|---|---|---|---|---|
| | | | **Initial Set** | **Final Set** | |
| 3000 | 360 | 28 | 180 240 | | 27.5 |

**Table 2.** Chemical properties of cement (quality score/%).

| SiO₂ | CaO | Al₂O₃ | MgO | SO₃ | Fe₂O₃ | Na₂O | Other |
|---|---|---|---|---|---|---|---|
| 21.13 | 62.35 | 4.58 | 1.78 | 3.25 | 4.25 | 0.22 | 2.44 |

The carbon steel used for making the sample is HPB335 ordinary round steel bar. The chemical composition is shown in Table 3. First, the steel bar is cut into a cylinder with diameter and height of 10 mm, and then the working face of the steel bar is polished step by step with 400#~1200# terrazzo sandpaper.

**Table 3.** Chemical properties of HPB335 ordinary round steel bar (w/%).

| C | Si | Mn | P | Al | Cr | Ni | Fe |
|---|---|---|---|---|---|---|---|
| 4.00 | 3.25 | 20.70 | 0.95 | 1.00 | 1.50 | 1.50 | 67.00 |

### 2.2. Preparation of the Multifunctional Rust Inhibitor

First, nitrite and phosphate were weighed according to the mass ratio of 5:1; the nitrite was 1000 g and the phosphate was 200 g. They were ground in a mortar for 5 min to obtain the antirust component. Then, fly ash and mineral powder, according to a certain proportion, were uniformly stirred to obtain anti-corrosion components. Finally, 240 g of the expansion component magnesium oxide (MgO) were added, mixed and stirred, taken out after 15 min, and sealed and stored for later use. The nitrite, phosphate, and MgO were purchased from Shanghai Aladdin Biochemical Technology Co., LTD (Shanghai, China). All the reagents used in the synthesis procedure are analytically pure grade chemicals.

### 2.3. Experimental Method

#### 2.3.1. Weight Loss Method

A weight loss test, as a simple and reliable method for metal corrosion measurement, is widely used in metal corrosion research. Polished steel bar samples were degreased with absorbent cotton soaked with water-free ethanol, scrubbed twice, and dried with a hair dryer. After drying, an analytical balance was used to weigh and record each sample. Then, the samples were soaked in 5% NaCl and simulated concrete hole solution (pH = 12.3 saturated calcium hydroxide solution), and the rust inhibitor was added at dosages of 0%, 2%, 4%, and 6%. After soaking for 7 d, the samples were taken out. After pickling, the samples were cleaned in distilled water, and the surface was cleaned with a brush.

After drying, the samples were weighed and recorded. The corrosion rate and inhibition efficiency (IE$_w$) according to the mass loss were calculated as follows:

$$v = \frac{w_0 - w}{st} \times 100\% \tag{1}$$

$$IE_W = \frac{v_0 - v_1}{v_0} \times 100\% \tag{2}$$

where $w_0$ and $w$ are the samples quality before and after testing, respectively; $v_0$ is the corrosion rate of the control samples; $v_1$ is the corrosion rate of the sample after adding rust inhibitor; $t$ is immersion time; and $s$ is exposed electrode area. The weight loss experiments were carried out with five parallel samples under every condition.

### 2.3.2. Compressive Strength Test of Concrete

After taking out the 60 mm × 60 mm × 60 mm concrete cube test block at the curing age, taking the unformed surface as the upper and lower compression surface, the center of the specimen was geometrically centered with the press, the loading speed was 0.6 MPa/s, and the failure load was recorded. The cube compressive strength formula is as follows:

$$F_{cu} = \alpha \frac{N}{A} \tag{3}$$

where $F_{cu}$ is the compressive strength of the concrete cube, $N$ is the failure load, $A$ is the compression area, $\alpha$ is the size conversion coefficient. The side length of the test block is 100 mm and the conversion coefficient is 0.95.

### 2.3.3. Chloride Ion Electromigration Experiment

The rapid chloride ion electromigration (RCM) method was selected for this experiment, according to GB/T 50082-2009, the "general concrete long-term performance and durability test method standard". The steel bars before embedding in concrete were pickled to remove rust, polished, cleaned with anhydrous ethanol, and finally dried with cold air. The concrete rods of size 40 mm × 40 mm × 160 mm were prepared and removed after 24 ± 2 h in the mold, and then immersed in a water tank for 28 d of curing. The anode solution was 10% NaCl solution and the cathode solution was 0.3 mol/L NaOH solution. The color indicator was 0.1 mol/L AgNO$_3$ solution. All reagents used to prepare the solution are analytically pure grade chemicals.

### 2.3.4. Carbonation Experiment

Cube specimens, 60 mm × 60 mm × 60 mm, under standard curing for 28 d, were used for the carbonation experiment. Each specimen was taken out one day before and placed in a drying box for 48 h. After drying, the two non-test surfaces were sealed with paraffin wax. Parallel lines were drawn at 10 mm spacing on the reserved test surface as measuring points of carbonation depth, and the test blocks were put into a rapid carbonation test box. The concentration of CO$_2$ in the box was controlled at 20 ± 5%, the temperature was controlled at 25 ± 2 °C, and the humidity was controlled at 70 ± 5%. When the carbonation time reached 7 d and 28 d, the specimens were taken out, and the cube specimens were split from the middle by the splitting method of a pressure testing machine. Then, 1% phenolic phthalein alcohol solution was sprayed on the section of concrete, and after waiting 30 s, the carbonation depth of each measuring point was measured using a steel ruler according to the pre-drawn 10 mm bisection line. The formula for calculating carbonation depth of concrete test blocks with different carbonation times is as follows:

$$d_t = \frac{1}{n} \sum_{i=1}^{n} d_i \tag{4}$$

where $d_t$ is the average carbonation depth of the specimen after carbonation t days, *n* is the total number of measurement points, and $d_i$ is the carbonation depth of each measuring point.

### 2.3.5. Electrochemical Test Method

A three-electrode system was used in the electrochemical experiment, in which the research electrode, the opposite electrode, and the reference electrode were carbon steel specimen, platinum electrode, and saturated calomel electrode, respectively. The carbon steel electrode was immersed in 3.5% NaCl solution until the corrosion potential reached stability, and then tested with a Princeton Versa STAT 4 electrochemical workstation.

### 2.3.6. Morphological Characterization

The carbon steel surface morphologies were characterized by scanning electron microscopy (SEM) (SU1510, Tokyo, Japan). The surface morphology features of the carbon steel were observed by atomic force microscopy (Agilent, AFM-5500, California, USA), and the surface morphology of the electrode was observed continuously in situ with tapping mode under OCP.

## 3. Results and Discussion

### 3.1. Weight Loss Analysis

Figure 1a shows the weight loss test results of the steel bar in 5% NaCl and saturated $Ca(OH)_2$ solution with different dosages of the multifunctional rust inhibitor. It can clearly be observed that the weight loss rate of the sample decreases gradually with an increase in the content of rust inhibitor. The weight loss rate of the sample without rust inhibitor was 0.23%, and the weight loss rate was reduced to 0.06% after the addition of rust inhibitor. This shows that it is difficult to inhibit the occurrence of rust in the environment without rust inhibitor, so the quality loss is greater. The change in rust inhibition efficiency with the amount of rust inhibitor is shown in Figure 1b. It can be observed from the figure that with an increase in the content of rust inhibitor, the rust inhibition efficiency increases gradually. When the mixing amount is 6%, the increased amplitude of the antirust efficiency tends to be gentle with an increase in the mixing amount. This may be due to the addition of sufficient rust inhibitor on the surface of the carbon steel, which forms a relatively uniform and dense protective film, effectively inhibiting rust in the samples. An increase in the content of antirust agent will not significantly improve the antirust effect.

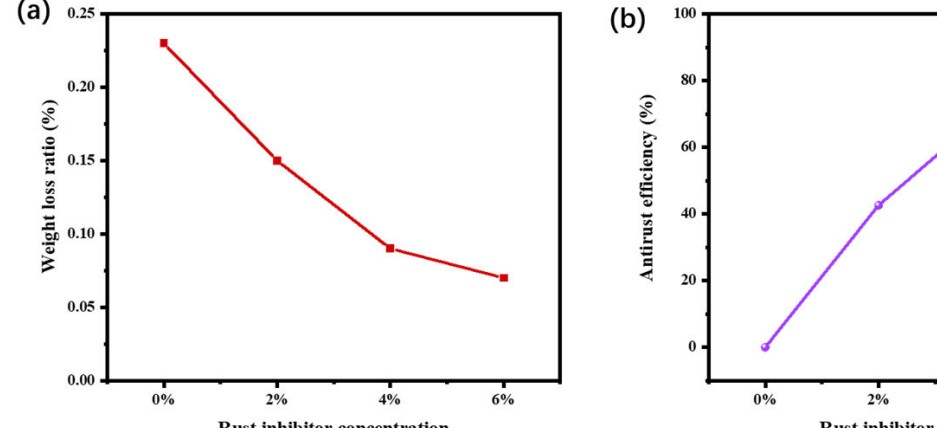

**Figure 1.** (**a**) The samples' weight loss ratio and (**b**) rust resistance efficiency change with the amount of rust inhibitor.

### 3.2. Micromorphology Analysis

In order to analyze the changes in the surface morphology of the samples after corrosion, SEM testing was conducted. Figure 2a shows the surface topography of the carbon steel before the corrosion test. It can be seen that there are several polishing marks of terrazzo sandpaper grinding. The sample morphologies after the corrosion experiment without and with corrosion inhibitors are shown in Figure 2b,c, respectively. The surface of the carbon steel sample without rust inhibitor appears rough, without any visible grinding scratches. However, pitting rust pits are present, indicating that the chloride ions have eroded and destroyed the passivation film on the carbon steel. The surface of the carbon steel soaked in the solution of 4% rust inhibitor is smooth without visible rust pits, and there is a granular substance adsorbed on the surface of the carbon steel. These granular substances may be rust inhibitor components, which can be adsorbed on the surface of the carbon steel, so as to form a layer of uniform and stable passivation film, effectively protecting the surface of the carbon steel and inhibiting the occurrence of corrosion [29].

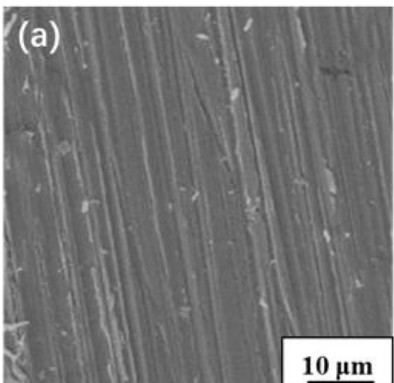 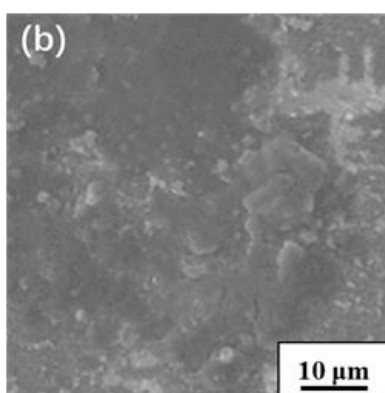 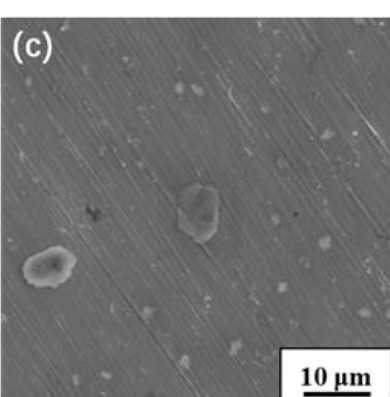

**Figure 2.** SEM microstructure of steel bar samples: (**a**) before the corrosion experiment; (**b**) without corrosion inhibitor, after the corrosion experiment; (**c**) with corrosion inhibitors, after the corrosion experiment.

In order to further explore surface changes in the samples after the corrosion experiment, AFM was used to test the surface morphology changes in the samples. The surface plan and 3D height diagram of carbon steel samples soaked after the corrosion experiment with and without corrosion inhibitors are shown in Figure 3. As can be seen from the planar graph in Figure 3a1, large rust spots formed on the surface of the carbon steel after corrosion, and the matrix stripes on the surface of the carbon steel are covered. By observing the 3D height diagram (Figure 3a2), it can be seen that the surface of the carbon steel is attached with uneven patches of embroidery well. As can be seen from the planar graph in Figure 3b1, it is obvious that the addition of rust inhibitor still makes the base fringe visible on the surface of the carbon steel, which is not covered by rust spots, and the 3D height map is smoother and more uniform than that without the addition of rust inhibitor. In addition, a roughness analysis of the height map shows that the surface roughness RIS of the carbon steel without rust inhibitor is 19.39, and that of the carbon steel with rust inhibitor is 12.07, indicating that the multifunctional rust inhibitor can make the passive film on the surface of carbon steel more flat, and can have a better protective effect on carbon steel. The analysis results show that the rust inhibitor is adsorbed effectively on the surface of carbon steel passivation film, which can improve the defects of passivation film.

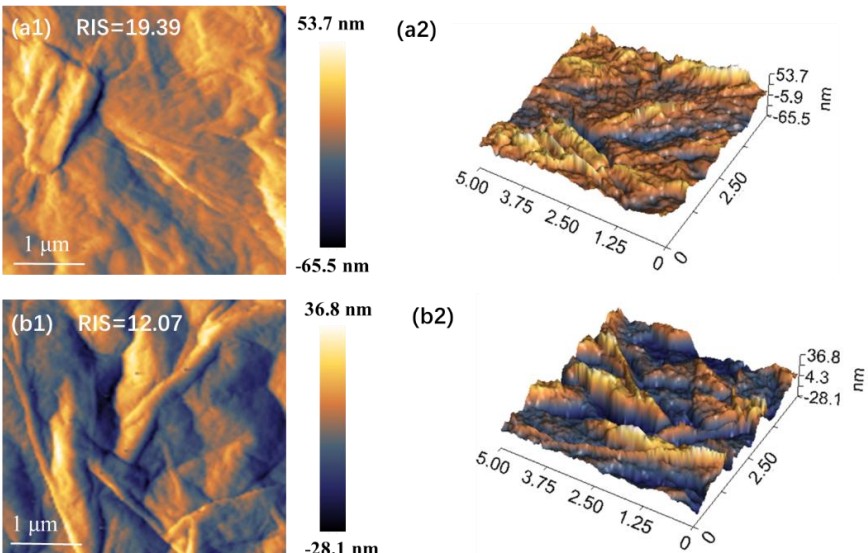

**Figure 3.** AFM images of carbon steel samples soaked after the corrosion experiment: Without corrosion inhibitor, (**a1**) surface plan and (**a2**) 3D height diagram; with corrosion inhibitors, (**b1**) surface plan and (**b2**) 3D height diagram.

### 3.3. Effect of Rust Inhibitor Mixed with Concrete Rods in Concrete

The influence of different multifunctional rust inhibitor addition amounts on rods in concrete compressive strength is shown in Figure 4a. As can be seen from the figure, the incorporation of the multifunctional rust inhibitor affects the early strength of the concrete. The 3 d and 7 d compressive strength of the concrete with rust inhibitor added is slightly lower than that of the rods in concrete without rust inhibitor, and the 3 d strength is reduced by about 2.5%. The compressive strength of the concrete mixed with rust inhibitor was higher than that of the control group at the curing age of 15 and 28 days. The compressive strength of the rods in concrete mixed with rust inhibitor increased significantly at the curing age of 4%, which was 11.8% higher than that of the blank group. The main reason is that the main component of the multifunctional rust inhibitor is mineral admixture. The early strength of concrete mainly comes from the hydration of cement. With an increase in the rust inhibitor admixture, the content of the mineral admixture increases, the content of cement decreases, and the early compressive strength of rods in the concrete decreases. With an increase in curing time, the activity of volcanic ash in the mineral admixture is gradually stimulated, which promotes the generation of hydration products, refines the pore size of concrete, and improves the compactness of the concrete [30]. The results demonstrate that the incorporation of a multifunctional rust inhibitor can effectively enhance the compressive strength of concrete rods, while exhibiting good adaptability to such rods and no adverse impact on their compressive strength.

Figure 4b shows the $Cl^-$ diffusion coefficient of 28 d rods in concrete. It can be observed that the incorporation of multifunctional rust inhibitor significantly reduces the chloride ion diffusion coefficient of rods in concrete and improves the chloride ion erosion resistance of concrete. The diffusion coefficient of chloride ion decreases with an increase in rust inhibitor content. When the dosage of rust inhibitor was 2%, the chloride ion diffusion coefficient decreased significantly, which was 41.2% lower than that of the control concrete. The diffusion coefficients of chloride ions at 4% and 6% of antirust agent decreased with an increase in the antirust agent content. The main reason may be that the composition of rust inhibitor promotes the hydration reaction [11], so that the hydration products in concrete increase, fill the internal pores, refine the pore structure, make the internal structure of concrete more dense, improve the impermeability, and thus inhibit the corrosion of steel bars. In addition, the antirust components in the multi-function antirust agent can play the role of curing and adsorption of chloride ions, preventing the chloride ions in the

environment from entering the specimen, thereby improving the durability of the rods in concrete.

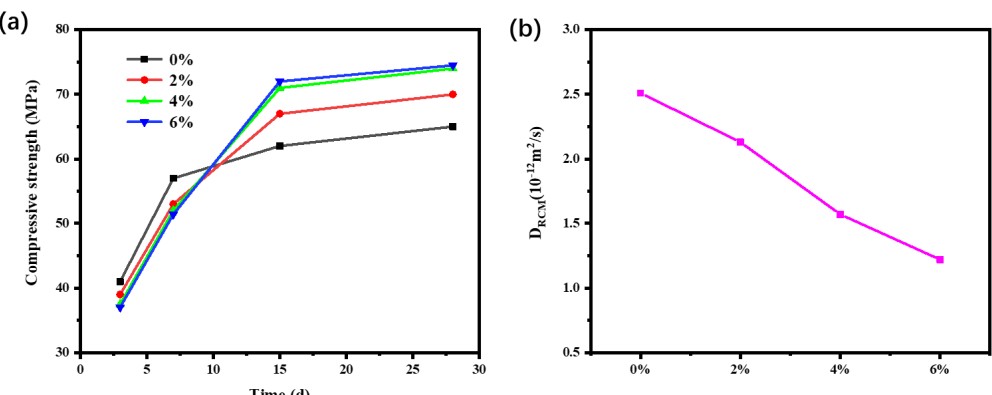

**Figure 4.** (**a**) Compressive strength of rods in concrete; (**b**) chloride ion diffusion coefficient of rods in concrete doped with different content of rust inhibitor.

### 3.4. Effect of Carbonation Resistance of Concrete

The concrete carbonation depth of different carbonation times is shown in Figure 5. Figure 4a shows the carbonation depth chart for 7 days of carbonation, and it can clearly be seen that the addition of the multifunctional rust inhibitor has no significant influence on the carbonation depth of concrete for 7 days. The analysis shows that the concrete has low water glue, strong carbonation resistance, and dense internal pores, so $CO_2$ fails to invade the concrete in a short time and has a significant influence on the pH value of the concrete. Figure 4b displays the carbonation depth chart for 28 days of carbonation, and shows that, with an increase in carbonation time, the effect of carbonation resistance becomes more and more obvious. It can be seen that the carbonation depth of concrete decreases significantly after adding rust inhibitor, and decreases with an increase in mixing amount. When the content of rust inhibitor was 4%, the 28 d carbonation depth decreased significantly by 21% compared with the control group. The results show that the addition of rust inhibitor can improve the carbonation resistance of concrete to some extent. The carbonation mechanism of concrete is that $CO_2$ diffuses into concrete channels and reacts with alkaline hydration products to reduce the internal alkalinity of concrete, thus causing corrosion of steel bars [31]. According to the analysis, due to the multifunctional compound rust inhibitor, nitrite promotes the hydration reaction, so that the hydration products in the concrete increase, and the complex dense components, blocking concrete channels, detailed concrete aperture, so that the concrete internal structure is more dense; carbonation is inhibited, and therfore analkaline environment is maintained inside the concrete and steel corrosion is inhibited.

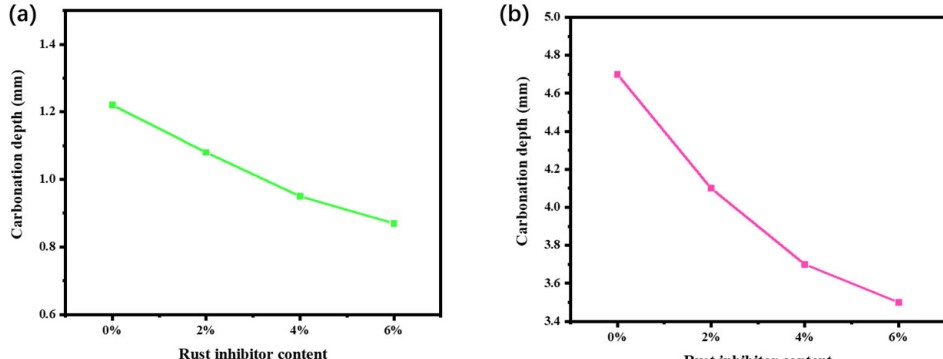

**Figure 5.** Carbonation depth of concrete with different dosage amounts of rust inhibitor: (**a**) Carbonation time is 7 d; (**b**) carbonation time is 28 d.

### 3.5. Electrochemical Test Analysis

The electrochemical impedance spectroscopy (EIS) of samples with different dosage amounts of rust inhibitors are shown in Figure 6a. The electrochemical impedance diagram has only one capacition-reactance arc, and the sample itself is equivalent to a resistor and a capacitor in parallel, so the EIS can be fitted using the circuit model shown in Figure 6b [32] (where Rs is the solution resistance, Qc is the corrosion product capacitance, and Rc is corrosion product resistance). It can be observed from the figure that the resistance of the carbon steel sample without adding rust inhibitor is 1500 Ω, and the resistance increases to 1800 after adding rust inhibitor. The larger the impedance value, the smaller the self-corrosion current density and the better the corrosion resistance of the samples, indicating that the corrosion resistance of the samples can be significantly improved by adding rust inhibitor.

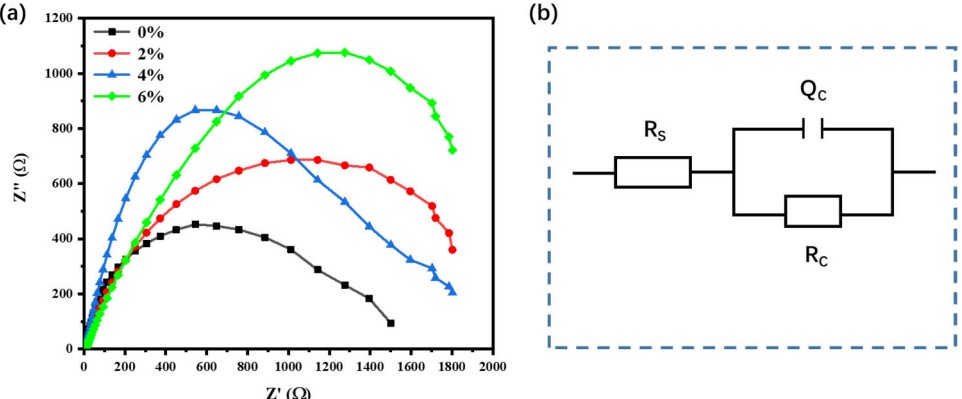

**Figure 6.** (**a**) EIS pattern; (**b**) equivalent circuit model of EIS.

### 4. Conclusions

In summary, in this study, we developed a novel multifunctional composite rust inhibitor that was specifically designed to enhance the corrosion resistance and durability of reinforced concrete. Through the corrosion experiment of carbon steel in simulated concrete hole solution with different dosages of rust inhibitor, the surface of carbon steel was tested by SEM and AFM to study the effect and mechanism of the rust inhibitor. In addition, the influence of the rust inhibitor on the performance of concrete was studied by combining the compressive strength, chloride ion electromobility, rapid carbonation ring tests, and electrochemical experiments under different rust inhibitor contents. Through the metal corrosion weight loss test, it was observed that the rust resistance efficiency gradually increased with an increase in the mixture of compound nitrite rust inhibitors. When the content of antirust agent is 4%, the antirust efficiency is 82.6%. When the content of antirust agent is greater than 4%, the increase in antirust efficiency tends to be gentle. At the same time, the measured electrochemical impedance of AC impedance spectrum is the largest, indicating that the concentration of the rust remover can effectively restrain the corrosion of the steel bar, and it has a good rust inhibition effect. After the SEM observation of the surface of the carbon steel, it can be seen that the addition of the rust inhibitor can effectively protect carbon steel. According to the observation of the surface of carbon steel by AFM, adding the rust inhibitor can significantly make the surface passivation film stable and smooth, and in the case of adding chlorine salt, it can also form a stable film. In addition, the larger the amount of multifunctional rust inhibitor, the better the performance of concrete. The addition of the rust inhibitor can reduce the porosity of concrete, and with an increase in the dosage, the pore structure is further refined, the compactness of concrete is enhanced, and the compressive strength, chloride ion penetration resistance, and carbonation resistance of concrete are improved. In subsequent research, more attention should be given to the combination with the actual environment, considering the effects of

harmful ions $SO_4^{2-}$ and $Cl^-$ on the durability and mechanical properties of concrete, as well as the influence on the reinforcement in concrete.

**Author Contributions:** Formal analysis, X.L.; Investigation, Z.N.; Data curation, Y.L.; Writing—original draft, X.L.; Writing—review & editing, Z.N.; Project administration, Z.N. and Y.L. All authors have read and agreed to the published version of the manuscript.

**Funding:** This research received no external funding.

**Institutional Review Board Statement:** Not applicable.

**Informed Consent Statement:** Not applicable.

**Data Availability Statement:** Not applicable.

**Conflicts of Interest:** The authors declare no conflict of interest.

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
