# Peer review of "The Effects of a Multifunctional Rust Inhibitor on the Rust Resistance Mechanism of Carbon Steel and the Properties of Concrete"

_coatings, doi:10.3390/coatings13081375_

Round 1
Reviewer 1 Report
Authors give interesting qualifications of one type of inhibitor for prevention of rust.
However, all experimental procedures for re-checking of results are not provided, and experiments would not be adequately confirmed.
Please add more details.
Furthermore, experiments are not performed directly on steel bars in concrete but indirectly by testing the effect on steel, and independently influence on concrete. For that reason, please give more discussion regarding the behavior of concrete with steel bars, and how your results transfers to real situation.
Do not cite author words from other literatures. Hearsay. Every author mentioned in the text needs to have original paper added to references.
Further observations are given in the attached document.

Author Response
1. The reference for the application has been modified accordingly.
2. Standard sand is the standard sand proposed by the International Organization for Standardization (ISO R679-68), the main component is SiO2.
3. The compressive strength of concrete is tested separately, and the steel bar is not inserted into the concrete.
4. What is tested in the experiment is the compressive strength. The mark in Figure 4 is wrong and has been modified.

Reviewer 2 Report
Comments to the authors
Article “Effect of Multifunctional Rust Inhibitor on Rust Resistance Mechanism of Carbon Steel and Properties of Concrete”
The Corrosion inhibitors are based on using chemical materials which are physically or chemically adsorbed over the metal surface and change the interface chemistry, forming a protective film on the surface, so the topic of the investigation is interesting, There are many investigations and proposals in the world to try to mitigate or reduce the problem of corrosion of reinforcing steel, therefore the comments on the work are the following:
The comments are as follows:
· Line 8: The word Summary is repeated twice, review.
· Line 8: The statement "To solve rebar corrosion in existing concrete structures" change the text because the results do not show that they have solved the problem of corrosion of reinforcing steel in concrete structures, what was shown is that its inhibitor has a percentage of effectiveness, in the conditions of the study variables, the corrosion of the steel until now has not been able to be resolved, which is why there is an immensity of work in the world.
· Line 11 and 12: Change the wording of "The results show that the weight loss of carbon steel with multifunctional rust inhibitor is reduced obviously and the rust inhibition efficiency is up to 82.6%", it is a scientific work, please write scientifically .
· Line 99: Change the wording of “The test results showed that the cement has good physical properties”, Indicate which standard or regulation the cement used complies with, scientific technical wording, avoid colloquial wording.
· Line 142: Indicate which standard was used for the test “2.3.4. Chloride ion electromigration experiment”
· Line 148: This test does not exist “2.3.5. Carbonization experiment”, must be the essay on Depth of Carbonation not Carbonization, review the literature that is very extensive on the subject. So you'll need to change carbonization to carbonation throughout the document.
· Line 167: Change the subtitle “2.3.6. Characterization technology”, It is physical chemical characterization or morphological characterization.
· Line 173: In the contribution “3.1. Weight loss analysis”, indicate what percentage of mass loss is critical to indicate that there is severe corrosion in reinforcing steel in concrete structures in aggressive environments.
· Lines 226 and 227: Refers to the compression test and indicates that it is figure 1, which in the document is on line 186 and is the weight loss figure, in addition to figure 4, which indicates that it is the compression sample results of tension not compression.
· In references: The format established in the authors' guide is not respected
· Please read your document carefully and order it. Improve scientific writing in the third person and analyze its results with other research. It is a work that, although it is interesting, does not explicitly indicate why they placed the inhibitor in the rod or in the pore solution or the dimensions of the concrete specimens for their compression tests and the chloride permeability test. Please show the dosage of the concrete, standards used, images of the specimens when they were made, when they were tested for compression and chloride permeability and carbonation, place the images of the carbonation profile.

Author Response
1. Duplicate has been deleted.
2. The corresponding sentence has been modified.
3. It is described in more scientific terms.
4. The reference standard of cement was added.
5. The test standard of chloride ion migration experiment was supplemented.
6. We have changed carbonization to carbonation in manuscript.
7. The title of 2.36 has been changed to morphological characterization.
8. The percentage of weight loss has been added to the manuscript.
9. This corresponds to Figure 4b.
10. We have perfected the scientific description of writing. The rust inhibitor is added to the slurry of the concrete to supplement the size of the test sample. The dosage and application standards of cement have been added, and the sample production process is a physical picture, which is not suitable for direct inclusion in the paper.

Reviewer 3 Report
The issue of corrosion of steel in concrete is very important. The presented article mainly deals with the corrosion of steel and its reduction by using a rust inhibitor that directly acts on steel. Separately, concrete samples subjected to CO2 – carbonation were examined, but it is not clear what was to result from it.
In the article, steel tests should be left and concrete tests should be discarded because they do not contribute anything to the experiment. It is necessary to focus only on the study of corrosion inhibitor steel, for the time being without reference to concrete, because it works differently than the solution in which the authors are examining. Applications should also be limited.
From the title discard "properties of concrete"
The mislabeled drawings in chapter 3.3 should be Figure 4a a is Figure 1 and should be Figure 4b and is Figure 2
For language correction
Author Response
- The carbonization mechanism of concrete is that CO2 diffuses into concrete pores and reacts with alkaline hydration products to reduce the internal alkalinity of concrete, thus causing corrosion of steel bars.
- Steel bar and concrete are usually used simultaneously in construction projects, and the purpose is to explore the suppression of steel bar rust in concrete structures. It is necessary to explore the effect of rust inhibitor on concrete.
-
The serial number of the figure has been changed accordingly.

Reviewer 4 Report
1. The abstract needs to be finalised. In the current state the annotation is not informative.
2. Not enough electrochemical studies. It is difficult to judge the protective properties of the investigated materials. It is necessary to include cyclic volt-ampere characteristics in the work.
3. For the obtained results: Figures 1 and 4 - confidence intervals are necessary, which will allow to judge about the reliability of the conducted research.
4. Scanning microscopy results with additional investigation of infrared spectroscopy data should be described in more detail.
Moderate editing of English language required
Author Response
- The abstract has been supplemented and modified.
2. Electrochemical experiments were performed to test electrochemical impedance spectroscopy of the samples.
3. Due to the limitation of experimental conditions, we did not test so many sets of data as the sample size to calculate the confidence interval.
4. The description of the scanning microscopy results with additional investigation of infrared spectroscopy data have been supplemented in more detail.

Round 2
Reviewer 2 Report
The comments were attended by the authors
Author Response
Dear reviewers, thank you for your careful review and constructivesuggestions regarding our manuscript.
Reviewer 3 Report
As already noted, rust inhibitors must be tested together with concrete, and not separately. The effect of corrosion of the solution and inhibitor on steel rods rather than rods in concrete was investigated. Minor fixesAuthor Response
Dear reviewer, thank you for reviewing our manuscript and for the constructive comments, which greatly helped us to improve the manuscript. In chloride ion electromigration experiment and electrochemical experiment, we tested effect of corrosion of the solution and inhibitor on steel rods.Due to the limitations of other experiments, only steel bars or concrete were selected. In the future research, we will redesign the experiment according to your suggestions and combine the experimental conditions.Dear reviewers, thank you for your careful review and constructivesuggestions regarding our manuscript. We have polished the English language and marked all the amends on our revisedmanuscript.

Reviewer 4 Report
Принять в настоящем виде
Author Response

(The authors gave the same response as above.)
